# Medicinal Characteristics of *Withania somnifera* L. in Colorectal Cancer Management

**DOI:** 10.3390/ph16070915

**Published:** 2023-06-22

**Authors:** John M. Macharia, Zsolt Káposztás, Raposa L. Bence

**Affiliations:** 1Doctoral School of Health Sciences, Faculty of Health Science, University of Pẻcs, Vörösmarty Mihály Str. 4, 7621 Pécs, Hungary; 2Faculty of Health Science, University of Pẻcs, 7621 Pécs, Hungarybence.raposa@etk.pte.hu (R.L.B.)

**Keywords:** *Withania somnifera* L., pharmacological activity, colorectal cancer, apoptosis, cytotoxicity, antiangiogenesis, antimigratory effects

## Abstract

Research into tumorigenic pathways can aid in the development of more efficient cancer therapies and provide insight into the physiological regulatory mechanisms employed by rapidly proliferating cancer cells. Due to the severe side effects of cancer chemotherapeutic medications, plant chemicals and their analogues are now explored more frequently for the treatment and prevention of colorectal cancer (CRC), opening the stage for new phytotherapeutic strategies that are considered effective and safe substitutes. Our study aimed to evaluate the medicinal properties of *Withania somnifera* L. and its safety applications in CRC management. Important databases were rigorously searched for relevant literature, and only 82 full-text publications matched the inclusion requirements from a massive collection of 10,002 titles and abstracts. *W. somnifera* L. contains a high concentration of active plant-based compounds. The pharmacological activity of the plant from our study has been demonstrated to exert antiproliferation, upregulation of apoptosis, decrease in oxidative stress, downregulation of cyclooxygenase-2 (COX-2), induction of targeted cytotoxic effects on cancerous cells, and exertion of both antiangiogenesis and antimigratory effects. We advise further research before recommending *W. somnifera* L. for clinical use to identify the optimal concentrations required to elicit beneficial effects in CRC management in humans, singly or in combination.

## 1. Introduction

### 1.1. Botanical Description and the Biodiversity of Withania somnifera (L.) Plant Species

*Withania somnifera* (L.) (*W. somnifera* (L.)) belongs to the kingdom Plantae (plants), subkingdom Tracheophytes (vascular plants), division Angiospermae, class Eudicots, clade Asterids, order Solanales, family Solanaceae, subfamily Solanoideae, tribe Physaleae, genus *Withania*, and species *somnifera*, according to the biological classification system [1,2,3]. *W. somnifera* L. is a Solanaceae plant commonly called “ashwagandha” in Hindi/Sanskrit and is a small shrub growing in abundance in the subtropics. The leaves and roots are used in Ayurveda, the Indian traditional medicine [4]. The aerial part, particularly the stem, leaves, and calyx, are sparsely covered with fine hairy tomentum [5]. It is an evergreen shrub growing to a height of about 1.5 m. They have simple, glabrous, ovate, petiolated, entire, shiny, smooth, and opposite leaves extending to a length of 10 cm. The branches are erect and are about 60–120 cm in length [5]. Flowers appear bright yellow or greenish and are approximately 1 cm in length. They have small fruits growing to a size of approximately 6 mm in diameter. The fruits grow like berries and turn orange-red when they are mature. Their seeds are yellow and about 2.5 mm in diameter [6], and are small, flat, yellow, reniform, and very light. The crop is generally grown in the Kharif season, and the plant has a taproot system, being 15–25 cm in length and light yellow in color (Figure 1). It requires dry weather conditions for the development of better root quality and alkaloid content [5].

Additionally, *W. somnifera* Dunal and *W. somnifera* Kaul have been identified as two subspecies [7]. In India, the *Withania* species *W. somnifera* and *W. coagulans* are both widely farmed [2]. There are two *W. somnifera* cultivars utilized in Sri Lanka, although the Indian cultivar is better suited for drug research due to its starchy character, whereas the local species has fibrous roots that are challenging to powder [7]. The leaves and roots of this medicinal plant species are used in Ayurveda, the Indian traditional medicine. In Ayurvedic, allopathic, Unani, homeopathic, and other medical systems, its roots, seeds, and leaves have been utilized for more than 3000 years for a variety of health-related purposes. It is often referred to as the Queen of Ayurveda or as a Rasayana plant because of its exceptional medicinal powers [2]. It is grown throughout the arid tropical regions of Afghanistan, Baluchistan, the Canary Islands, China, Congo, Egypt, Israel, Jordan, Madagascar, Morocco, Nepal, Pakistan, South and East Africa, Spain, Sri Lanka, Sudan, and Yemen [8]. The traditional medical system of Ayurveda has a long history and is widely regarded. To achieve its therapeutic goals, this method makes use of numerous natural compounds in diverse forms [9]. The Ayurvedic system identifies thousands of plants that are helpful in preventing ailments and maintaining health, including *Withania somnifera* L. [9].

**Figure 1 pharmaceuticals-16-00915-f001:**
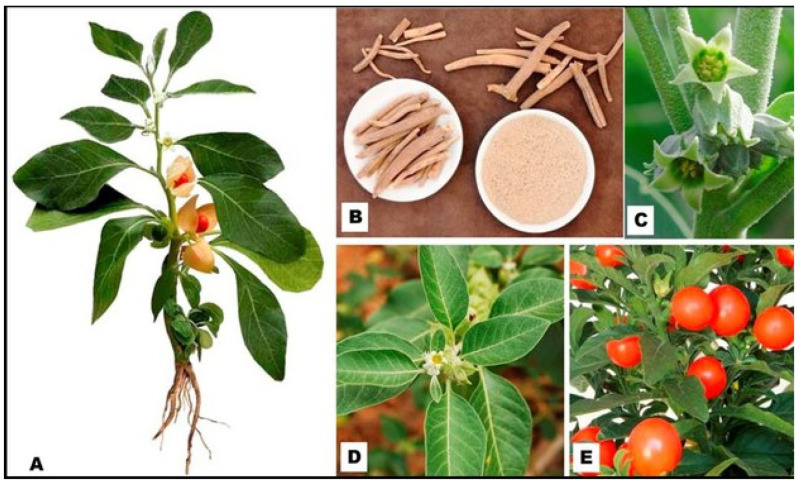
*Withania somnifera* L. plant and its specific organs: (**A**) the main plant, (**B**) the roots and their ground root powder, (**C**) the flowers, (**D**) the leaves, and (**E**) the fruits [10].

### 1.2. The Global Incidence of Colon Cancer (CRC) and Its Risk Factors

Cancer is the second-largest cause of death, trailing only cardiovascular disease. Colorectal cancer is now the third-biggest cause of cancer-related deaths in Western nations [11]. Cancer kills more than 70% of people in developing countries [12]. Even though many CRCs are intermittent, a significant percentage (5–6%) have a clear genetic relationship. The most frequent kind of gastrointestinal (GI) tract cancer is CRC [13]. Diets low in whole grains, vegetables, and fruits; high-fat diets; gender; race; age; family history; and region all have a role in the high prevalence of colon cancer. Lack of exercise, smoking, and excessive alcohol use have also been identified as significant contributing factors [14]. Exposure to infectious agents and heavy metals, such as lead, have also been identified as significant risk factors for CRC [13].

### 1.3. Future Strategies for Long-Term CRC Management and Contemporary Treatments

Understanding tumorigenic pathways can help researchers create more effective cancer treatments and gain insight into the metabolic control mechanisms used by swiftly reproducing malignant cells [15]. Even though the pathophysiology of colorectal cancer (CRC) is unclear and complicated, associations among risk factors appear to be essential to the emergence and progression of the illness [16]. 

The extracellular matrix, which includes the stromal and immune cells it surrounds and the chemical compounds and signaling substances in the intercellular space, constitute the tumor microenvironment (TME) [17]. A TME must be developed for the growth of cells, their migration, invasion, and metastasis to increase from the primary tumor location [18]. Due to the high frequency of cancer metastases, detecting adenomatous polyps, which are pre-CRC abrasions for colon cancer, is a common foundation for CRC strategies for prevention [19]. Lactic acid bacteria (LAB) prebiotics and probiotics have gained popularity over the last decade as a way to modify the tumor microenvironment (TME) [20,21].

Due to the severe side effects of these chemotherapeutic medications, plant chemicals and their analogues are now being used more frequently in both the treatment and the prevention of CRC [22]. Chemotherapy has significant side effects that emerge after healthy, normal cells become damaged [23], which opens the door for new phytotherapeutic strategies as effective and secure substitutes. Indirectly or directly, plant-based natural products account for around 50% of all regularly used chemotherapeutic therapies, making them an abundant source of novel anti-colon-cancer medications. It is important to highlight that a number of the pharmaceuticals available today were made from plant sources, which are used as a reliable source of medications [24]. In developing nations, the medical and healthcare sectors still heavily rely on plants and products made from them. The use of medicinal plants (herbs) to cure illnesses is nearly widespread in nonindustrialized cultures and is frequently more affordable than utilizing pricey standard drugs [25]. To determine and obtain significant benefits from these medicinal plant species, ethnopharmacological studies should be encouraged to investigate the most effective method of plant selection, extraction, chemical synthesis, and dose calculation [26].

Despite its lengthy history of therapeutic uses, its medicinal benefit has garnered less attention. The technique that *W. somnifera* (L.) uses has not been fully understood and established [2]. Our study aimed to evaluate the medicinal properties of *Withania somnifera* L. and its safety application in CRC management. This updated review presents the plant as a viable alternative with favorable benefits in CRC therapeutic management. As it is, however, there is little information available on this significant plant species, particularly in in vivo and in vitro CRC experimental trials. Therefore, this is the first comprehensive review evaluating the proactive potential for *W. somnifera* (L.) in the treatment of CRC in humans.

## 2. Methods

### 2.1. Applied Study Design and Electronic Databases

This study used an in-depth systematic review technique to assemble data on botanical descriptions, ethnomedical uses, plant-based compounds, and pharmacological activities of *W. somnifera* species of plants against CRC. To identify primary studies examining the effects of *W. somnifera* on CRC carcinogenesis, relevant literature was comprehensively searched in the Web of Science, Ovid, BMC, Springer, Elsevier, MDPI, and MEDLINE databases. Google Scholar was useful in acquiring more articles. This exercise was conducted in adherence to PRISMA guidelines [27] but with minimal modifications (Figure 2). The legitimacy of the Google Scholar papers was established by comparing them with the pertinent publication [28,29]. 

### 2.2. Review Question and the Screening Criterion

To make the search criteria easier, review questions were created. Two specific questions were set up: “Are the active compounds of *Withania somnifera* L. efficacious in colon cancer management?” “Are active compounds of *Withania somnifera* L. safe for human therapeutic application?” After setting up the review questions, the authors individually examined only English research titles and abstracts from primary sources [30,31]. Research focusing on the small intestine mucosa and research relevant to illnesses other than CRC were specifically omitted from the analysis. After being deemed eligible by five authors, titles and abstracts that met the basic satisfactory conditions were selected for full-text article assessment and subsequently used to provide the essential statistical data for the present review. To reduce the possibility of prejudice, authors’ independence was required while deciding whether to use the enrolled articles [30,31]. Only 82 full-text articles out of a sizable collection of 10,002 titles and abstracts met the inclusion standards (Figure 2).

### 2.3. Effective Search Strategy

A systematic search criterion was created to ensure that the screening method was effective. The MeSH terms “growth”, “proliferation”, “development”, “anti-cancer”, “bioactivity”, “phytochemicals”, “physiological activity”, “pharmacological activities”, and “multiplication” were combined with either “colorectal”, “colon cancer”, “adenocarcinomas”, “polyps”, “colorectal tumor”, “colon tumor”, keywords, or their synonyms. These terms were ultimately combined with the plant species under review, *Withania somnifera*.

## 3. Bioactive Compounds Present in *W. somnifera* L.

*Withania*’s biochemistry has been thoroughly investigated, and several distinct metabolites have been identified and their biological activities described (Table 1). Whereas withanolides (steroidal lactones that are highly oxygenated C-28 phytochemicals) are also isolated from other families, the *Withania* genus is a significant source of steroidal lactones in nature (Figure 3). Taccaceae, Solanaceae, and Fabaceae are some of these families [32]. From various portions of *W. somnifera* L., steroidal lactones and phenolic acids are two important chemical families that have been discovered [33]. 

Withanolide A, withaferin A (Figure 4 and Figure 5), isopelletierine, anferine, sitoindosides VII and VIII, sitoindosides XI and X, withanoside V, withanolide B, and withanoside IV are the most physiologically active botanical constituents of several species of the *Withania* genus [34]. Vanillic acid, p-coumaric acid, syringic acid, benzoic acid, gallic acid, physagulin, and trigonelline are only found in the leaves, whereas palmitic acid, oleic acid, linoleic acid, linolenic acid, withanone, 5,6-epoxy withaferin-A, 27-deoxywithanone, 27-hydroxywithanone, withanolide B, withanoside IV, withanoside VI, (+)-catechin, and 12-deoxywithastromonolide are isolated from both leaves and roots [33] (Table 1). 

### 3.1. Mechanisms of Carcinogenesis and the Inhibitive Potential Role of W. somnifera L. in CRC Growth and Proliferation

Through gene silencing and pathway analysis, Widolo et al. [50] reported that *W. somnifera* L. and its constituent parts kill cancer cells via at least five separate routes, including p53 signaling, GM-CSF signaling, death receptor signaling, apoptosis signaling, and the G2/M DNA damage regulation pathway. The most typical signaling was p53. It has been reported that tumor necrosis factor alpha (TNF-α), interleukin (IL)-10, and glutathione (GSH) levels are all lowered by aqueous root extracts of *W. somnifera* L. (0.05–0.4 mg/mL), which also regulate peripheral blood mononuclear cell (PBMC) and leukemic THP-1 cell viability [51].

Withaferin A suppressed IL-6, COX-2, TNF-α, and prosurvival biomarkers (NOTCH1, NF-κB, and p-Akt) in APC Min/+ and dextran sodium sulfate/azoxymethane models and prevented CRC in both models. The treatment of withaferin A reduced cancer proliferation in a transgenic adenomatous polyposis coli (APCMin/+) mice model [52]. 

In the CRC cell lines SW-480, SW-620, and HCT-116, withaferin A suppressed Notch-1 and inhibited the Akt/NF-B/Bcl-2 pathway. In colon cancer cells, withaferin A inhibited the rapamycin signaling factors p4E BP1 and pS6K and stimulated c-Jun-NH2-kinase-mediated apoptosis [53]. It has been reported that withaferin A triggered G2/M arrest in HCT116 and SW480 cancer cell lines [54]. Withaferin A also caused Mad2 and Cdc20 deterioration, in addition to mitotic delay, by disrupting the spindle assembly checkpoint mechanism. In a xenograft mice carcinoma model, Choi and Kim [55] reported that withaferin A suppressed the development of HCT116 human CRC cells. The outcome was mainly accomplished by inhibiting STAT3 transcriptional activity. The effect of *W. somnifera* L. extract on HT-29 colon cells was investigated. The investigators found that methanolic extract of *W. somnifera* L. root had an antiproliferative effect due to increased ROS generation and mitochondrial dysfunction [9].

Alnuqaydan et al. [56] investigated the promotion of antiproliferative effects by 5-fluorouracil (5-FU) and withaferin A. In colonic cells, the combined therapy suppressed the β-catenin pathway, which has been linked with cell cycle arrest during the G2/M phase. According to the findings, the combination of withaferin A and 5-FU promotes cell viability via ER stress-mediated activation. A four-week therapy of *W. somnifera* (L.) extract has been demonstrated to improve immunological dysfunction and CRC rehabilitation [57]. The immunomodulatory action was found to be beneficial for the management of CRC. Changes in leukocytes, neutrophils, lymphocytes, immune system complexes, and immunoglobulin A (IgA), IgG, and IgM antibodies induced the impact [57]. The downregulatory potential of 27-desoxy-24, 25-dihydrowithaneferin A, 27-O-glucopyranosylviscosalactone B, withaferin A, withanolide sulfoxide, and withanoside IV has been reported with decreased expression of COX-2 [52,58,59] and Bcl-2 [52] in CRC. Withaferin A exposure has been demonstrated to enhance the amount of late apoptotic cells and the aggregation of cells in the subG1 arrest in the cell cycle. Furthermore, it inhibits the levels of antiapoptotic proteins, such as Bcl-2 and Bcl-xl, while inducing apoptosis via PARP and caspase-3 cleavage [52,53,54,60] (Table 2). 

### 3.2. Cytotoxicity and Safety Properties of Withania somnifera L.

When treating oncologic disorders, the impact of anticancer medications on healthy cells is a major source of worry. Researchers have undertaken very little research to fully grasp this significant problem. Withanolides have been investigated for their potential cytotoxicity toward normal cells as well as their powerful anticancer properties [9], including radiosensitization, immunomodulation, anti-inflammation, antimetastatic, and antiangiogenic effects [62]. 

A combination of withanolide A and withaferin A offers multimodal activities against CRC cell lines. These chemical components are reported to trigger the cytotoxicity of cells in various cancerous cell lines [40]. The proposed cytotoxicity mechanisms comprise activating the extrinsic and intrinsic apoptosis signal cascades, induced by the intensified generation of nitric oxide reactive and oxygen species within the cancerous cells [13]. The standardized extract of *W. somnifera* (L.) did not exhibit any toxicity at a concentration of 2000 mg/kg, according to a specific acute and subacute toxicity investigation [9,63]. Another comparable in vivo investigation [64] reported no harmful effects from the hydroalcoholic extract of *W. somnifera* L. roots. The *W. somnifera* L. root extract did not have any maternal or fetal toxicity in mice, according to an investigation on gestational developmental adverse effects [65]. 

According to research by Baig et al. [66], withanolides have a lower cytotoxic effect on normal cells than they do on cancerous cells. For example, with an IC50 of 99.7 g/mL, the extract exhibited cytotoxic action against lung cancer cells [67]. In a different investigation, *W. somnifera* L. extract’s antiproliferative efficacy additionally demonstrated lower cytotoxicity in normal cells as compared with DU145 prostate cancer cells [57]. Similar findings were made regarding the cytotoxic activity of *W. somnifera* L. extract on normal Vero cells compared with the human MDA-MB-231 breast carcinoma cell line [68]. Withaferin A had negligible cytotoxic properties on normal fibroblasts (TIG-1 and KD cells) but had anticancer properties on prostate cancer cells (PC-3 and DU-145). Withaferin A and withanone therapy in combination have shown improved cytotoxic ability against tumor cells. The restriction of growth and invasion in the cancerous cells was brought about by the downregulation of hnRNP-K, VEGF, and MMP-3 [62]. 

*Withania* and its numerous species have also been examined for their harmful effects [69]. According to 69 clinical and preclinical studies, *Withania somnifera* L. roots were shown to be effective and safe. It has been demonstrated to be nontoxic and effective clinically for human health and wellness, from its use in antiquity to its present applications [70].

### 3.3. Significance of Targeted Upregulated Apoptotic Activity in CRC Management

Apoptosis is a mechanism that has undergone little change throughout evolution and is crucial for maintaining tissue homeostasis [71]. The process of apoptosis requires specially designed machinery. A proteolytic mechanism that uses a family of proteases known as caspases is the main part of this machinery [72]. The death-receptor-mediated mechanism and the mitochondrial pathway have been identified as two separate but convergent pathways for caspase activation. 

The mitochondrial caspase pathway’s starter, caspase-9, is a crucial mediator in the control of apoptosis. These enzymes (caspases) are part of a cascade that is initiated by proapoptotic commands, which leads to the cleavage of an assortment of peptides and the disintegration of the cell. The comprehension of caspase programming is essential for strategically regulating apoptosis for therapeutic benefit [71,72]. The suppression of natural apoptosis has been hypothesized to enhance the incidence of cancer [73,74]. Similarly, it has been noted that a higher prevalence of colorectal adenoma is highly correlated with a lower rate of apoptosis [75]. The capacity to trigger apoptosis in epithelial cells of gastrointestinal origin is one of the potential strategies in chemoprevention [74]. As a result, investigating the apoptotic mechanism is a viable avenue for CRC. When determining the prognosis for individuals with stage II CRC, the degree of expression of the apoptosis-associated genes caspase-9 and caspase-10 could prove useful. It appears that the carcinogenesis of CRC involves both the death-receptor-mediated and mitochondrial pathways. The life span of aberrant mucosa cells is increased by the decreased expression of caspase-9 and caspase-10 [76,77]. As a result, these cells have the potential to undergo further gene mutations and eventually give rise to cancerous cells. 

It has been suggested that the activity of withanolides is mediated via the control of nuclear factor kappa B (NF-kB) expression because NF-kB regulates numerous genes that affect cell division, malignancy, cancer metastases, and inflammation. This suggests that withanolides prevent NF-kB activation and NF-kB-regulated gene expression, which could help clarify why they can increase apoptosis while preventing invasion [78]. According to a study using the leukemic murine mouse model, withanolide D lowers anti-apoptotic genes (TERT, Bcl-2, and Puma) [51]. By virtue of high levels of ROS, the dysregulation of Bax/Bcl-2 expression, and the concurrent disruption of mitochondrial membrane potential (ΔΨm), a novel fraction of proteins isolated from *W. somnifera* L. roots could stimulate mitochondria-mediated apoptosis in triple-negative breast cancer cells (MDA-MB-231) at an IC50 dose of 92 g/mL. Reported also [79] were caspase-3 stimulation, G2/M cell cycle arrest, and nuclear lamin protein cleavage. Furthermore, the crude water extract (0.5%) of WS modified the signaling cascade, including proapoptotic and tumor-promoting proteins, which helped to inhibit tumor growth [51]. 

The association between caspase-9 and CRC is still poorly understood to date. Studying its association with clinicopathological characteristics and longevity may provide insightful data for predicting survival and choosing additional treatment strategies. *W. somnifera* L. ethanolic extract is cytotoxic (99.7 g/mL) on cancerous cells, induces apoptosis, and inhibits angiogenesis and cell migration [67]. The possible anticancer action of the extracts of *W. somnifera* L. has been shown to be due to the increased autophagy induction and apoptotic effects of the plant. Withaferin A causes apoptosis and Mad2 and Cdc20, crucial components of the spindle checkpoint complex, which are degraded by proteasomes. By restoring correct anaphase initiation and maintaining a greater number of viable cells, further overexpression of Mad2 partially reverses the harmful effect of withaferin A. It is hypothesized that withaferin A kills cancer cells by delaying the mitotic exit and then causing chromosome instability [54]. Although various *W. somnifera* L. compositions have highly promising anticancer properties in both in vitro and in vivo applications, there are currently no authorized therapeutic candidates. For the discovery of novel anticancer pharmaceuticals, conducting clinical trials with *W. somnifera* L. phytochemicals/formulations is urgently necessary.

### 3.4. Antiangiogenic and Antimigratory Potential of W. somnifera L.

Angiogenesis is essential for the survival, development, and metastasis of cancer cells. Sajida [67] demonstrated that the genes and proteins involved in tumor angiogenesis, such as VEGF, angiogenin, and MMP-2, were downregulated, inhibiting angiogenesis. Some ingredients derived from medicinal plants have demonstrated strong antiangiogenic activity. Conferone caused oxidative stress and cell suicide by inhibiting angiogenesis in HT-29 CRC cells through an altered secretome that included vascular endothelial growth factor, angiopoietin-1, and two factors [80]. 

Additionally, withaferin A demonstrated antiangiogenic efficacy in the mouse model with reduced p-ERK and p-Akt levels. By reducing the expression of MMP in HeLa and PC3 cells, this molecule also prevented the invasion of cancer cells [81]. At lower concentrations, *W. somnifera* L. extract has been implicated with significantly reducing cell migration, thus demonstrating its antimigratory ability [67]. It has thus been reported to have antimetastatic potential since it significantly reduces MMP expression. Another investigation utilizing bioinformatics and biochemical strategies revealed that phytochemicals present in *Withania somnifera* L. decreased the levels of the migration-promoting proteins hnRNP-K, VEGF, and MMP, making them potential alternative therapies for the management of metastatic cancer [62]. 

It has further been elucidated that withanone and withaferin A in combination had powerful antimigratory and antiangiogenic effects in vitro and were specifically lethal to cancer cells. These actions were corroborated by molecular studies of marker proteins, such as MMPs, which are essential for the invasion and metastasis of cancer [50,62,82]. Thus, it may be drawn that the antimetastatic activity is achieved by targeting multifunctional RNA-binding protein, hnRNP-K, and withanone-rich combination of withanone and withaferin A restricts cancer cell migration, and angiogenesis in vitro and in vivo.

## 4. Conclusions

*Withania somnifera* L.’s pharmacological activity has been shown in our study to unquestionably exert antiproliferation, upregulation of apoptosis, decrease in oxidative stress, downregulation of cyclooxygenase-2 (COX-2), induction of targeted cytotoxic effects on cancerous cells, and exertion of both antiangiogenesis and antimigratory effects. Numerous macromolecules found in *W. somnifera* L. suggest the value of using this plant as a possible anti-CRC agent with the guarantee of phytotherapeutic benefits. *W. somnifera* L. and its constituent parts kill cancer cells via at least five separate routes, including p53 signaling, GM-CSF signaling, death receptor signaling, apoptosis signaling, and the G2/M DNA damage regulation pathway, as has been reported. However, due to the lack of established quantities and concentration measurements, we advise further research before recommending *W. somnifera* L. for clinical use to identify the minimal and optimal concentrations/levels required to elicit beneficial effects in the management of CRC in humans, either alone or in combination. Evidence adduced herein demonstrates this plant as an efficacious and safe substitute for human medicinal developments. Finally, it may also be investigated as prospective raw materials for the formulation of conventional CRC pharmaceuticals.

## Figures and Tables

**Figure 2 pharmaceuticals-16-00915-f002:**
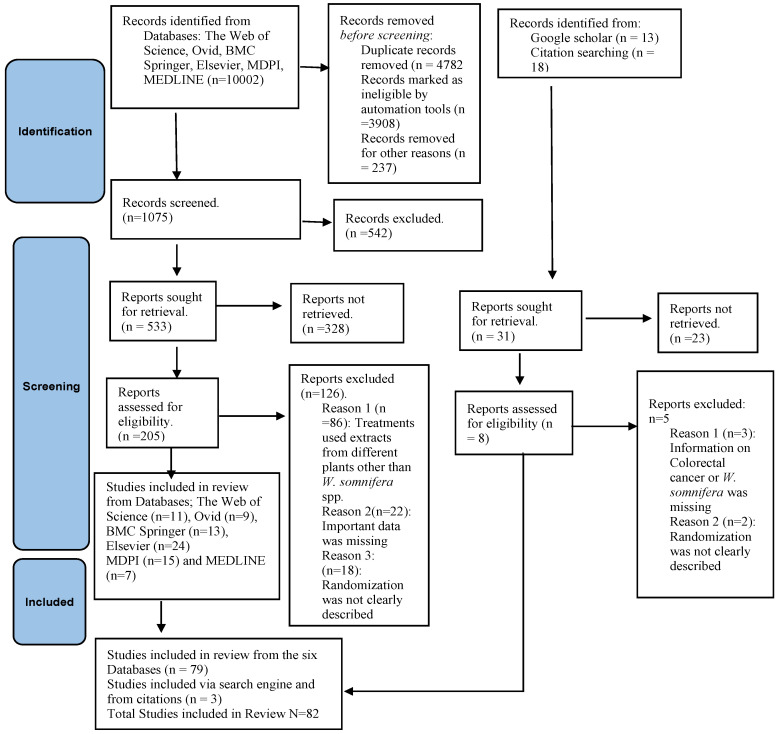
Adopted PRISMA flow diagram with slight modifications for the implemented systematic review scheme.

**Figure 3 pharmaceuticals-16-00915-f003:**
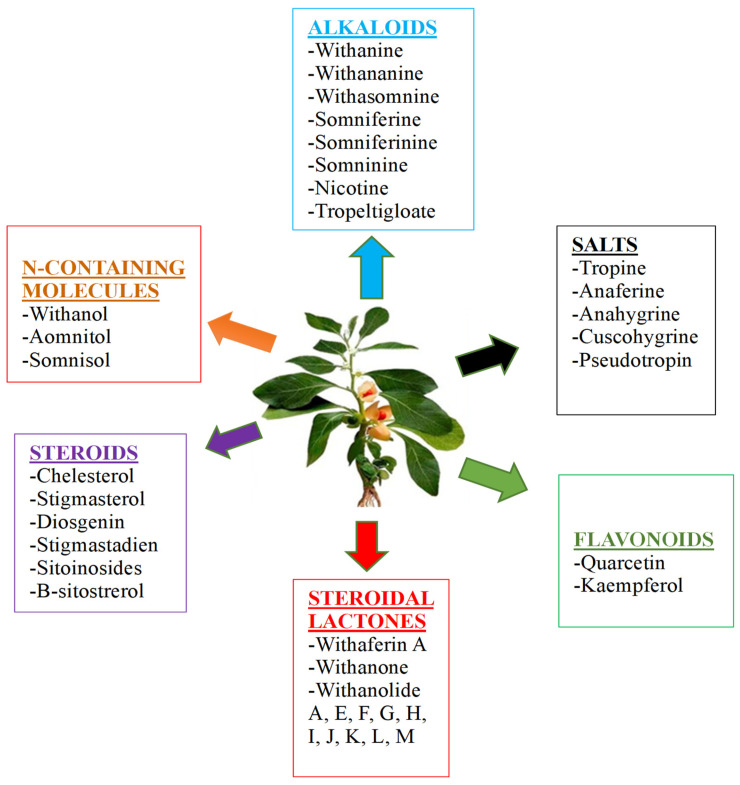
Classification of the various metabolites present in *W. somnifera* L.

**Figure 4 pharmaceuticals-16-00915-f004:**
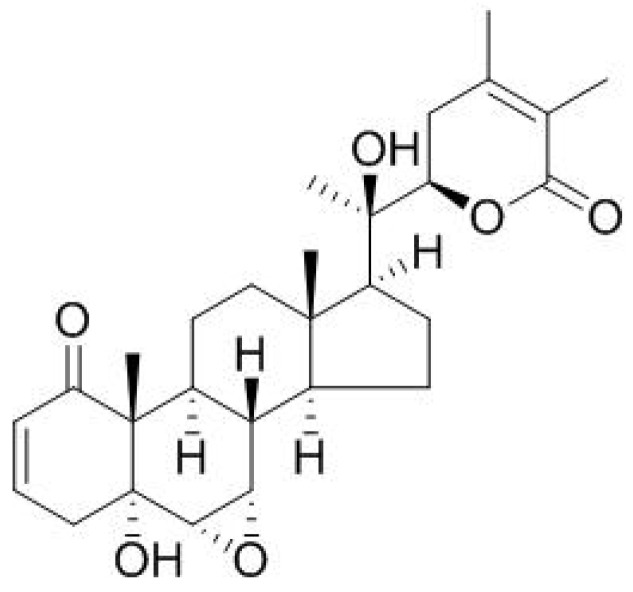
The chemical structure of withanolide A.

**Figure 5 pharmaceuticals-16-00915-f005:**
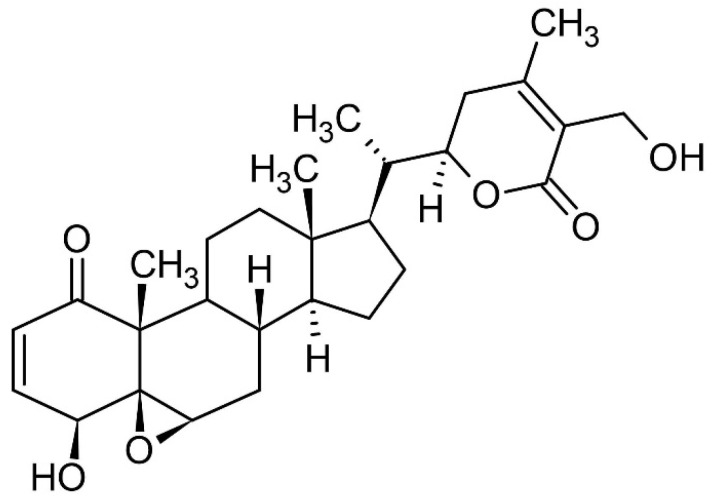
The chemical structure of withaferin A.

**Table 1 pharmaceuticals-16-00915-t001:** Active metabolites present in *Withania somnifera* L. and their pharmacological activities.

Plant Part/Organ.	Bioactive Compounds	Pharmacological Activity	References
Roots	Withanolide A, withaferin A, withanolide sulfoxide, withanoside IV and VI, withacoagin, dihydrowithanolide D, ixocarpalactone A, glucopyranosyl moieties, glycosides, withasomidienone, 5,7 α-Epoxy-6α, 20α-dihydroxy-1-oxowitha-2, 24-dienolide, ß-sitosterol, ß-sitosterol glucoside, stigmasterol glucoside, Viscosa lactone B, 16ß-acetoxy-17(20)-ene, 6a-hydroxy-5,7a-epoxy, 27 hydroxy withanone, hydroxy, 17-deoxy withaferin A, deoxy withastromonolide, physagulin, benzyl alcohol, 2-phenyl ethanol, benzoic acid, p-hydroxy, phenyl acetic acid, asparagine, choline, palmitic acid, oleic acid, linoleic acid, porphyrine	Anti-inflammatory, memory enhancement, cerebellar ataxia, antiperoxidative, cardiotonic, and antioxidative abilities; also enhances fertility	[2], [32,33,34,35,36,37,38,39,40,41,42,43,44,45,46,47]
Leaves	Withanolides, withaferin, polyphenols, saponins, alkaloids, steroids, steroidal lactones, flavonoids, glycosides, 27 hydroxy withanone, hydroxy, 17-deoxy withaferin A, deoxy withastromonolide, physagulin, ß-sitosterol, ß-sitosterol glucoside, 2-hydroxy propanol, 2-hydroxy propanoic acid, 1-octanol, benzoic acid, butanedioic acid, phenyl acetic acid, p-hydroxy, phenyl ethano, p-hydroxy benzoic acid, alanine, aspartate, asparagine, choline, palmitic acid, oleic acid, linoleic acid, porphyrine, pheophytin, sterol, TAG, vanillic acid, p-coumaric acid, syringic acid, gallic acid, physagulin, and trigonelline	Antistress, antianxiety, anticarcinogenic activity, antimicrobial, antioxidative ability, anthelmintic, and anti-inflammatory potency	[2] [32,33,34,35,36,37,38,39,40,41,42,43,44,45,46,47]
Fruits and flowers	Chamase, condensed tannins, peroxidases, proteolytic enzyme, cystine, flavonoids, glutamic acid, amino acids, aspartic acid, alanine glycine, hydroxyproline, isopsoralen, psoralen, proline, tyrosine, and valine	Antimicrobial activity, management of respiratory illness	[2,46,47]
Stems	Alkaloids and its derivatives (ashwagandhine, isopelletierine, pseudotropine, [3]-tigloyloxtropine, tropeltigloate, isopelletierine, hygrine, mesoanaferine, choline, somniferine, withanine, withananine, hentriacontane, visamine, withasomnine, somniferinine, somninine, nicotine, cuscohygrine, pseudotropin, anahygrine, anaferine, tropine), glycosides and its derivatives (withanosides I–VII, withanamide), flavonoids (quercetin, 7-hydroxyflavone, kaempferol), and phenolics and their derivatives (coumaric acid, caffeic acid, chlorogenic acid, gallic acid, ferulic acid, catechin)	Used to treat tumors, nocturnal leg cramps, coronary heart diseases, diarrhea, and psychiatric palpitation; improves blood cholesterol levels; antimicrobial; relaxant; antispasmodic; sedative; muscle relaxant; diuretic; strengthens capillary walls, osteoporosis	[2,46,47,48,49]

**Table 2 pharmaceuticals-16-00915-t002:** Anticolorectal cancer activities of *W. somnifera*’s bioactive constituents and their associated mechanisms of action.

Biological Mechanism	Bioactive Constituents	References
Downregulation of COX-2 (↓COX-2)	27-Desoxy-24, 25-dihydrowithaneferin A, 27-O-glucopyranosylviscosalactone B, 4,16-dihydroxy-5 h, 6h-epoxyphysagulin D, diacetylwithaferin A, physagulin D (1→6)-h-D-glucopyranosyl-(1→4)-h-D- glucopyranoside, viscosalactone B, withaferin A, withanolide sulfoxide, withanoside IV	[52,58,59]
Downregulation of NF-κB and PI3K/Akt (↓NF-κB, and PI3K/Akt)	Withaferin A, withanolide sulfoxide	[52,53,54]
Downregulation of Bcl-2 (↓Bcl-2)	Withaferin A	[52,53,54]
Upregulation of apoptosis (↑apoptosis)	Withaferin A, withanolide D, withanone, 4β-hydroxywithanolide E,	[52,60,61]
Upregulation of apoptotic caspase-3 gene (↑caspase-3)	Withaferin A	[52,60]

## Data Availability

No new data were created or analyzed in this study. Data sharing is not applicable to this article.

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
