# Peer review of "Medicinal Characteristics of Withania somnifera L. in Colorectal Cancer Management"

_pharmaceuticals, 2023, doi:10.3390/ph16070915_

Round 1
Reviewer 1 Report
Macharia et al have written a review article entitled “Medicinal characteristics of Withania somnifera L. in colorectal cancer management”. Authors have described the concept to a greater extent but the quality of manuscript is very low in terms of experimental study and quality is very low. Some major issues suggested here like,
· Not a novel concept, authors designed the concept in a very broad manner. It should be more specific either in terms of mechanistic approach or selected bioactive compounds.
· manuscript is not formatted properly,
· very limited experimental evidences,
· Figure 1 numbering repeated two times
· most of the sections are repeated content described already in the previous literature.
· Authors are suggested to remove the unrelated content and reformat the manuscript in a proper manner with more figure emphasized on the anticancer mechanism of bioactive compounds and content related tables.
· Authors are encouraged to re-submit the manuscript after resolving these issues.
· At this stage manuscript is not in the state of further consideration.
Thus, this manuscript cannot be considered for publication.
Moderate editing of English language required
Reviewer 2 Report
This study contributes to the field of cancer research by exploring the potential of Withania somnifera L. as a safe and effective alternative for the management of colorectal cancer. This research contributes to the development of new phytotherapeutic strategies for CRC and emphasizes the importance of conducting additional studies before recommending W. somnifera L. for clinical use in humans. There are certain points that need careful attention of the authors to address:
1. What databases were used in the literature search to identify primary studies examining the effects of Withania somnifera on colorectal cancer carcinogenesis and microbial growth?
2. How were the Google Scholar papers validated in this study?
3. What were the specific review questions set up in this study regarding the efficacy and safety of active compounds of Withania somnifera in colon cancer management?
4. What criteria were used to exclude certain research titles and abstracts from the analysis?
5. How many full-text articles met the inclusion standards after the screening process?
6. What are the bioactive compounds present in Withania somnifera L.?
7. Which metabolites have been identified in Withania somnifera, and what are their biological activities?
8. What are the physiological active botanical constituents of several species of the Withania genus?
9. What are the cytotoxic effects and properties of Withaferin A, withanone, and the alcoholic extract of Withania somnifera leaves on cancer cells?
10. What are the potential mechanisms through which Withania somnifera L. inhibits CRC growth and proliferation?
11. How does Withaferin A affect the expression of pro-survival biomarkers in colorectal cancer models?
12. What signaling pathways are involved in the cytotoxic effects of Withaferin A in colorectal cancer cell lines?
13. How does Withaferin A induce G2/M arrest in HCT116 and SW480 cancer cell lines?
14. What effect does Withaferin A have on the development of HCT116 human colon cancer cells in a xenograft mice carcinoma model?
15. What was the antiproliferative effect observed in HT-29 colon cells treated with Withania somnifera extract, and what were the underlying mechanisms?
Minor editing of English language required
Round 2
Reviewer 1 Report
Authors have revised the manuscript to a greater extent however I would like to draw the attention of authors towards bioavailability of the Withania somnifera compounds. Authors are suggested to add some content related to bioavailability of compounds of Withania in specific CRC models. After that manuscript can be considered for publication after careful evaluation.
Minor English and grammatical changes are required.
Author Response
Thank you very much for your thorough review of our previous round of submissions. We have added more content on the bioavailability of compounds of W. somnifera associated with CRC, as you have correctly advised and guided. In addition, Section 3.1 as highlighted, describes the demonstrated effects of W. somnifera active metabolites on various CRC models. Figure 2 summarizes the mechanisms of action exhibited by different bioactive ingredients present in W. somnifera.
Reviewer 2 Report
I have thoroughly reviewed the current version of this manuscript and find it to be highly suitable for publication. The authors have diligently addressed all previous concerns and made significant improvements that have substantially enhanced the overall quality and coherence of their work. The manuscript now exhibits a commendable level of clarity, precision, and scientific rigor, rendering it both valuable and impactful within its respective field. Consequently, I have no further comments or reservations and wholeheartedly recommend this manuscript for publication.
Author Response
Thank you very much for your wonderful comments and highly impactful level of thorough review. We are so grateful to you and the time you took in assessing and reviewing our work.